# ICT and Women’s Health: An Examination of the Impact of ICT on Maternal Health in SADC States

**DOI:** 10.3390/healthcare10050802

**Published:** 2022-04-26

**Authors:** Courage Mlambo, Kin Sibanda, Bhekabantu Ntshangase, Bongekile Mvuyana

**Affiliations:** 1Public Administration and Economics, Faculty of Management Sciences, Mangosuthu University, Umlazi 4031, South Africa; miriam@mut.ac.za (B.N.); bmvuyana@mut.ac.za (B.M.); 2Department of Economics, Faculty of Management Sciences, Walter Sisulu University, Mthatha 5099, South Africa; kinsibanda@wsu.ac.za

**Keywords:** maternal health, maternal mortality, ICT, eHealth, women’s health, development

## Abstract

Attainment of sexual and reproductive health is regarded as a human rights matter. Notwithstanding this, maternal mortality continues to be a major public health concern in low-income countries, especially those in sub-Saharan Africa. Maternal mortality remains high in Africa, yet there are information communication technologies (ICTs) (such as the internet, mobile communication, social media, and community radios) that have the potential to make a difference. Making effective use of all of these ICTs can considerably decrease preventable maternal deaths. ICTs, particularly mobile devices, offer a platform for access to health information and services that can bring change in areas where health infrastructure and resources are often limited. However, for Southern Africa, maternal mortality remains high despite the presence of ICT tools that have transformative potential to improve maternal health. In light of this, this study sought to examine the impact of ICT on maternal health. The study was quantitative in nature, and it used panel data that covered the period from 2000–2018. The Mean Group and Pooled Mean Group cointegration techniques and a generalised method of moments panel technique were used for estimation purposes. Results showed that ICT has a negative effect on maternal health. This shows that ICT tools contribute positively to maternal health. The study gave a number of recommendations. The mobile gender gap should be closed (digital inclusion), mobile network connectivity boosted, and digital platforms must be created in order to enhance the transformative potential of ICT in improving health outcomes.

## 1. Introduction

Attainment of sexual and reproductive health is regarded as a human rights matter. Despite this, maternal mortality remains a significant public health problem in low-income nations, especially those in sub-Saharan Africa. Sub-Saharan African countries suffer from the highest maternal mortality ratio (MMR) of 533 maternal deaths per 100,000 live births, or 200,000 maternal deaths a year [1,2]. Failure to ameliorate sexual and reproductive health has unfavourable outcomes as it also connects to various health outcomes such as infant mortality rates. Maternal mortality remains high in Africa, yet there are information communication technologies (ICTs) (such as the internet, mobile communication, social media, community radios) that have the potential to make a difference. Making effective use of all of these information communication technologies (ICTs) can considerably decrease preventable maternal deaths [3,4]. ICT interpositions provide an efficacious approach to alleviate maternal deaths and improve health outcomes.

Access to communication is widely recognized as an important factor in increasing the use and quality of maternal health care [5,6]. Across the globe, ICTs disrupt industries by drastically altering the means by which people engage with their wellbeing. The advancement of ICT may be leveraged as a game-changing communication tool to solve health issues. ICTs, particularly mobile devices, provide a platform for accessing health information and services, which can help to bring about change in locations where health infrastructure and resources are typically scarce [7]. Health information technology has the potential to become an effective instrument when used as part of a well-coordinated national health policy. It has the potential to empower policymakers, health frontline workers, and patients in all nations to enhance the quality of mother and child healthcare [8].

The prospects for mobile phones to act synergistically with existing health systems in developing countries are numerous [9,10]. By empowering individuals and providing decision makers with timely information about crucial health concerns, ICTs play a critical role in addressing health-related challenges. Because it is compact, portable, frequently utilized, and relatively affordable, the mobile phone has high potential, and expanding network coverage makes it easier to communicate with people in rural and remote locations [5]. Mobile platforms can allow health personnel to more proficiently monitor pregnancies, identify danger signs, and guarantee that women have access to emergency obstetric care in the event of problems [7]. Pregnant women can also receive mass health messages via their cell phones. Maternal care, chronic illness management, and disease epidemics can all benefit from mobile devices and mobile health (mHealth) services. These services increase the health system’s efficiency and effectiveness by tracking and reporting on patients, and they expand health care to disadvantaged populations [4,11]. People can use health management applications to check their health, keep track of their medications, and allow healthcare practitioners to review and share their medical information remotely [12].

ICT is valuable in data collecting, as well as the management of healthcare services, and has the potential to refine the quality and dependability of healthcare information and services [13]. It appears as a cost-effective and technologically feasible alternative to manually paper-based patient record keeping, ensuring that medical practitioners have access to the appropriate patient data at the appropriate time [14]. ICT can thus make a significant difference by improving access to information and healthcare services, as well as reaching out to women with information that will help them avoid avoidable deaths and problems. Clearly, ICT has a crucial role in the alteration of healthcare services. The use of ICTs as service and information delivery tools and platforms is, therefore, expected to enhance collaboration and efficiency in the health sector by disseminating health data and information and facilitating communication and collaboration [15]. Effective use of ICT in health can be a productive way to reduce health care costs and ameliorate the quality of care. Analysts point to the increase in the use of health apps and growing mobile connectivity as signs of an impending boom in the e-health sector in Africa [16]. Mobile technology is creating unparalleled opportunities for ameliorating maternal health outcomes.

Countries in Africa spend significant amounts of their GDP on delivering health services through systems that are often ineffective, expensive, and lacking in transparency. Information and communication technologies (ICTs) have the potential to alter how health services are delivered across Africa, not only in terms of efficiency, but also accountability [17]. The intake of ICT in health has not been explored in African health literature. This study argues that in order to order exploit the benefits of ICT for maternal health care, a thorough apprehension of the significance of ICT tools, particularly mobile phones, is imperative to improve viability, suitability and appraisal of such interventions [18]. It is thus worthwhile to examine the impact of ICT on maternal mortality in Southern African Development Countries (SADC).

## 2. Literature Review

### 2.1. Empirical Literature

The capacity of information and communication technology (ICT) to be integrated into present health institutions has been widely acknowledged in the literature [19,20,21,22]. There are numerous ways in which ICT can assist in improving maternal health. The potential of ICTs to generate, store, retrieve, and convey information among users can ameliorate and sustain the provision of healthcare solutions [23,24]. ICT, especially mobile phones, may be a proper channel for remedying the challenges of health literacy and for reaching women from inadequately serviced societies [21]. Mobile applications can provide patients with tools to enable their engagement in the treatment and welfare [25]. They can support the training of health personnel by providing access to correct and up-to-date information concerning health conditions and treatment as well as the most recent thoughts on treating certain health problems. 

Mobile technology permits patients to receive personal reminders via e-mail, automated phone calls, or text messages. Text messaging, for instance, exhibits significant potential as a means for health care improvement for a number of reasons; it is accessible on nearly every model of mobile phone, the cost is reasonably low, its use is extensive, it does not need substantial technological knowledge, and it is generally appropriate to a variety of health behaviours and circumstances [26]. There are cases where SMSs have proven to be effective in facilitating and improving health care activities. For example, in South Africa, Dis-Chem’s (a private health facility) decision to send SMS reminders dramatically reduced the no-show rate for patients who had registered for COVID vaccines. Dis-Chem used the SMS system to send reminders to people who had already been assigned an appointment [27]. (Khan, 2021). Mobile health improves doctor–patient engagement by permitting computer-generated consultation in real time using methods from text messaging to video calls. It also allows the patient to participate daily in their healthcare. 

Mobile technologies have been seen to be useful in improving maternal care. Mobile systems enable health care providers to track pregnancies more effectively, recognize danger indicators more quickly, and guarantee that pregnant women have easy access to emergency obstetric treatment if complications occur [28]. Oluwaseun (2021) [18] found that using ICT to care for pregnant women during and after delivery boosted demand for health services and reduced maternal-infant fatalities. Appointment reminders, health tip communication, and emergency referrals all benefit from ICT technologies (mobile phones, the internet, television/digital video disk (DVD), and radio). Déglise et al. (2012) [29] reviewed case studies from low-income countries to examine SMS-supported interventions for prevention, observation, administration and treatment compliance of communicable and non-communicable diseases. Their study showed that mobile phones seemed to be an effective instrument for disease control interventions in low-income countries. The study concluded that mobile phones function as a low-cost technique to address certain needs of health structures and can offer new opportunities to improve health care in low-income countries. 

Mimbi and Bankole’s (2015) [30] study indicated that countries that excelled in ICT inputs also excelled in their health systems. ICT also enhances life expectancy at birth and lowers infant mortality rates. African countries must spend heavily in ICT in order to enhance their health systems and achieve socioeconomic growth [30]. Majeed and Khan (2018) [22] conducted a study that examined the relationship between information and communication technology (ICT) and population health in 184 countries. The study used panel data and it showed that there was a positive relationship between ICT and population health. Ogechi and Olayini (2018) [31] examined the relationship and causality between ICT and health in Africa. The empirical findings revealed that information and communication technology (ICT) has a positive and statistically significant link with health, indicating that the greater the level of ICT usage, the better the health of the people. There was also a bidirectional causality between ICT and health. Obasola (2021) [32] examined the experiences of health facilities using ICT for maternal care in Nigeria. The study showed that using ICT increased the demand for health services.

### 2.2. Case Studies 

Case studies demonstrate how ICTs can improve maternal and healthcare in general. A scheme based on radio technology in providing maternal care to pregnant women adopted in Uganda in 1996 and brought about a 50% decrease in maternal mortality by the year 1999 [33]. A program called Mobiles4Health was created in Bangladesh with the goal of contacting health facilities to request midwives in the event that working women required them. The use of this approach resulted in an 89% increase in the number of births under the care of health experts. Earlier, approximately 90% of births took place outside health institutions [34]. In the US, there was an initiative called Text4baby. Text4baby was a nationwide health SMS service that provided timely information to pregnant women and new mothers in order to assist them improve their own and their newborns’ health. The Text4baby initiative’s merits were that it was a simple concept that made use of a widely used and familiar technology that was well integrated into everyday life [21].

In Indonesia, a pilot was initiated to see how mobile technology could help midwives working in remote regions. The goal of the initiative was to employ mobile technology to enhance the quality of health care in rural regions while also lowering maternal and newborn mortality rates. It also sought to improve communication between midwives and doctors and evaluate the viability of mobile data gathering. This program permitted midwives to remotely obtain medical information, gather patient information and vital health indicators, monitor patients, and actively contribute to this data sharing system [35]. In Uganda, a program called MobileVRS was launched. This program used a system that permits health team members to record important statistics via mobile phones. The system did not require software to be installed on mobile phones and it was compatible with any mobile phone. Costs related to data transfer were insignificant, and the project significantly lessened the amount of time needed to report important statistic events [35].

In Rwanda, a program called RapidSMS was initiated to improve maternal health by tracking pregnant women and newborns [28]. The mobile technology system created automatic reminders for appointments, delivery, and maternal-care-related visits. The aim was to improve increase attendance at antenatal care and postnatal care visits. Other proposed outcomes included the delivery of a fast link to emergency obstetric care and the establishment of a database of maternal records. The system also had a “Red Alert” mechanism which triggered emergency reaction and readiness by the receiving health institution, which contributed to real time management and therefore contributed to improving maternal health care [36].

In Zanzibar, a programme called Wired Mothers was launched to improve maternal care and other maternity-related problems [37]. The programme linked pregnant women to a primary health care facility through the use of mobile technology. The programme made use of text messages to remind pregnant mothers about appointments. The Wired Mothers initiative considerably improved the proportion of women receiving maternal care during pregnancy and after delivery. Apps for mobile phones have shown to be effective in enhancing the quality of mother and newborn health care [37,38]. In Tanzania, m-mama, an emergency transport system utilising mobile technology, was launched in order to reduce maternal and neonatal mortality rates. The programme provided a 24 h operating call centre to link pregnant women who experience maternal complications to an ambulance. The program also involved local private taxis who were trained to handle the transportation of obstetric emergencies [39,40,41]. (Touch Foundation, 2017; GMSA, 2018; Vodafone, 2021). Table 1 shows additional eHealth projects that are currently operational in selected developing countries. 

Table 1 shows that there has been a number of eHealth projects that are operational in selected developing countries. According to the World Health Organization (2021) [43], 42 of the 54 African nations have a national eHealth plan. These projects have been considered as having a good impact on health care. While the increasing number of eHealth policies across Africa indicates increased political desire, their presence does not ensure successful implementation [44]. The next subsection discusses some of the limitations that are faced by governments in adopting and implementing ICT systems (eHealth) in the health sector. 

### 2.3. Limitations

The analysis above showed that the ICT tools for maternal care service delivery can significantly improve maternal health. However, it has been seen that health facilities face challenges when using ICT tools for health service delivery. The challenges include poor internet connectivity, costs related to licensing and subscription, weak technical support, lack of expertise on how to use ICT systems, the rural–urban divide, poor network coverage, and lack of training on mobile applications [10,45,46]. The “implementation” duties and problems in poor nations are a major difficulty in maximising the advantages of ICT. Some people do not have access to the information and communication technology (ICT) system. In this scenario, ICT may exacerbate existing socioeconomic health inequities [47]. Likewise, some studies have criticized the adoption of ICT for improving health care. For instance, Bend (2004) [48] presents minor evidence to support the idea that ICTs can have an impact on public health. According to Krasovec [49] (2004), research revealed only shaky empirical data on the real impact of communication networks and suggested that access to communication technologies is not the answer to reducing maternal mortality in remote places. Because a woman needs emergency obstetric care in a short amount of time (due to significant bleeding, for example), excellent services must be available quickly and supported by good infrastructure management.

Gender disparities can also limit the use of ICTs for health: women have less access to mobile phones than men [50] and in certain communities, women’s traditional roles are at odds with the independence and professional responsibilities that mobile phones may provide [35]. It is for this reason that UNICEF (2009) [51] argued that examining the influence of mobile phones on maternal health necessitates investigating how mobile phones may be linked to the core cause of poor maternal health, which is women’s social status. The learning curve for utilizing mobile phones is steepened by high rates of illiteracy. Some people do not know how to operate some mobile applications. In some areas, issues of reliable access to electricity are a challenge [35,52]. A study by Obasola (2021) [32] showed that the challenges that are faced with using ICT in health care are unreliable power supply, additional costs associated with using ICT, and irrelevant calls. It can thus be said that ICT can undoubtedly help save lives on its own, but it is not a panacea. The effectiveness of any mHealth initiative depends on the overall health system being strengthened [53]. Noordam (2011) [5] concurs and argues that research is needed to develop the evidence base for scaling up mHealth and enabling informed mHealth policymaking, as well as to assess its value in ensuring timely delivery of medical equipment, providing health education, and improving access to reproductive health services such as family planning.

## 3. Materials and Methods

### 3.1. Data Sources 

The empirical analysis is based upon a panel data set. This study covers 15 SADC countries over the period from 2000–2018. The criterion for choosing this study period is the consistent availability of data for the period from 2000 to 2018. Data for all variables was sourced from the World Bank. 

### 3.2. Model Specification 

Majeed and Khan (2018) [22] analysed the relationship between information and communication technology (ICT) and population health. The analysis was based on econometric model of population health in 184 countries using panel data spanning from 1990–2014. This study builds on Majeed and Khan (2018) [22] and formulates the following model:(1)$${\mathrm{MM}}_{it}=\beta_{0}+\beta_{1}{\mathrm{HIV}}_{it}+\beta_{2}{\mathrm{ICT}}_{it}+\beta_{3}\mathrm{FR}+\beta_{4}{\mathrm{GDP}}_{it}+\beta_{5}{\mathrm{LE}}_{it}+\varepsilon_{it}$$
where MM is maternal mortality, HIV is the female HIV prevalence rate, ICT is total mobile subscriptions, FR is fertility rate, GDP is gross domestic product, LE is life expectancy, and $\varepsilon_{it}$ is an error term. The description of the variable presented in Equation (1) above is presented in Table 2 below.

The choice of variables was guided by literature. For instance, HIV was chosen because HIV-infected pregnant women have two to 10 times the risk of dying during pregnancy and the postpartum period compared with uninfected pregnant women [54]. Calvert et al. also argued that there is strong evidence that HIV increased the rate of direct maternal mortality. Fertility rate was chosen because fertility is an important determinant of maternal mortality, with high fertility levels associated with high maternal mortality [55]. GDP was chosen because economic development leads to improved health for both women and children through advances in the field of medicine [56].

### 3.3. Estimation Techniques

The estimation procedure involves four stages. The first stage is to examine whether the variables contain a unit root and the second stage is to examine whether there is a long run cointegrating relationship between the variables. The last stage estimates the short- and long-run coefficients. 

#### 3.3.1. Unit Root and Cointegration 

The stationary process of the series lies at the basis of the cointegration analyses developed for the determination of long-run relationships between time series and variables [57]. Therefore, unit root tests for time series were first performed. In this regard, two different types of panel unit root tests were employed to determine the order of integration between all the series in our dataset: Lm, Pesaran & Shin; and Levin, Lin & Chu. Then, a Pedroni panel residual cointegration test was performed to determine the cointegration relation between the variables. This test takes into account heterogeneity by using specific parameters that are allowed to vary across individual members of the sample and which consist of seven statistics for testing panel data cointegration [58]. Lastly, the panel ARDL model was performed in the last stage to examine the short-run and long-run coefficients.

#### 3.3.2. PMG and MG

This study used a panel ARDL approach. The ARDL (*p*, *q*) model is specified by the following equation:$$Y_{it}=\sum_{j=1}^{p}\varphi_{i,\;j}Y_{i,\;t-j}+\sum_{j=0}^{q}\delta_{i,\;j}X_{i,\;t-j}+\vartheta_{i}+\varepsilon_{it}$$

With *i* = 1, 2, …, N is the number of countries; *t* = 1, …, T is the time; *j* is the number of lags; *X_i_*_,*t*_ is the vector of the variables relating to maternal mortality and *ϑ_i_* is the specific fixed effect of the countries [59]. To consider the adjustment coefficient and the long-run dynamics, Equation (1) is reparametrized as follows:$$\Delta Y_{it}=\emptyset_{i}\;\left(Y_{i,\;t-1}-\emptyset_{i}\;X_{i,\;t}\right)+\overset{p-1}{\underset{j=1}{\sum }}{\varphi^{\prime }}_{i,\;j}\;\Delta Y_{i,\;t-j}+\overset{q-1}{\underset{j=0}{\sum }}{\delta^{\prime }}_{i,\;j}\;\Delta X_{i,\;t-j}+\vartheta_{i}+\varepsilon_{it}$$
where $\emptyset_{i}$ is the adjustment coefficient of the long run dynamics. $\emptyset_{i}\;$ indicates the long-run equilibrium relationship between $Y_{i,t}$ and $X_{i,t}$ ${\varphi^{\prime }}_{i,j}$ and ${\delta^{\prime }}_{i,j}$ represent the short run coefficients linking maternal mortality with its past values and the variables of interest $X_{i,t}$. A long-run association between maternal mortality and ICT is said to be present if $\emptyset_{i}$ is negative and significant. To estimate the ARDL model, two estimation techniques were employed, namely the mean group estimator (MG) and the pooled mean group estimator (PMG). According to Wang and Wang (2020) [60], the difference between these two estimators is that the MG estimator seems to be more consistent under the assumption that both the slope and intercepts are allowed to vary across countries, whereas the PMG estimator is consistent under the assumption of long-run slope homogeneity. The Hausman test was applied to identify the efficiency and consistency of each estimator over another. In other words, the Hausman test was used to choose the appropriate estimator. 

## 4. Result

### 4.1. Descriptive Statistics 

Before evaluating the data series, it’s crucial to look at the descriptive statistics to see how the variables vary and how they’re distributed [61] (Olayungbo and Quadri, 2019). The descriptive statistics of the variables employed in this investigation are shown in Table 3.

Results displayed in Table 3 show that GDP was positively skewed, that it had a value of 3.53, and that the other variables had a value closer to 0. GDP had a high kurtosis value (15.06) and other variables had kurtosis values that were closer to 3. Lastly, the Jarque–Bera statistic is below the 0.05% level of significance for all the series. This is an indication that the variables do not follow a normal distribution. 

### 4.2. Unit Root

The Levin, Lin, and Chu and the Lm, Pesaran, and Shin unit root tests were used in the study. The outcomes of these two tests are shown in Table 4 below.

Results from the Levin test showed that four variables had a unit root. These were the MM, LE, HIV, and GDP variables. They became stationary after being differenced once. LE and FR were stationary at levels. The Lm, Pesaran test also confirmed these results, but the only difference was in the HIV variable. The HIV variable was stationary at levels. Since the variables had different orders of integration, the next step was to check for cointegration. 

### 4.3. Cointegration 

Having found that the variables in question are integrated of order one I(1) and I(0), the next step was to test whether a long-run relationship exists between them. Table 5 shows the results from the Pedroni cointegration test.

Results show that out of the four panel statistics, three tests (v-statistic, rho statistic, and ADF statistic) show that there is cointegration amongst the variables. Under the group statistics, two group statistics (Group panel rho statistic, and Group ADF statistic) show that there is cointegration among the variables. It can thus be said that the Pedroni test finds that there is a long-run association between the variables under investigation. 

### 4.4. Presentation and Discussion of Main Results 

The paper estimates the effects of ICT on maternal mortality in the SADC region. The empirical results for the model are presented in Table 6. 

The results from both the MG and PMG appeared to be similar, but the strengths of the coefficients differed. Since there was no significant difference between the MG and PMG estimators, both results from both models will be discussed. 

Results show that ICT has a weak and negative relationship with maternal mortality. The coefficients of ICT of −0.075008 and −0.095005 are negative and statistically significant at the 0.51% level of significance for the MG and PMG estimates, respectively. This may support the claim that mobile technology is creating unparalleled opportunities for ameliorating maternal health outcomes. ICT, especially mobile phones, may be a proper channel for remedying the challenges of health literacy and for reaching women from inadequately serviced societies [21]. Mobile applications can provide patients with tools to enable their engagement in treatment and welfare [25]. Empirical evidence also supports the significant impact of ICT on maternal health. Nyamawe and Seif (2014) [38] link the decrease of maternal mortality in Tanzania between 1996 and 2010 to numerous projects that linked pregnant women to both health facilities and midwives using SMSs and calls. Obasola (2021) [32] examined the experiences of health facilities using ICT for maternal care in Nigeria. The study showed that using ICT increased the demand for health services. However, the relationship is marginal, and this may be ascribed to the challenges faced by African governments in creating a conducive environment for ICT-related programmes. Health facilities face challenges when using ICT tools for health service delivery. These challenges include poor internet connectivity, costs related to licensing and subscription, weak technical support, lack of expertise on how to use ICT systems, the rural–urban divide, poor network coverage, and lack of training on mobile applications [10,45].

The fertility rate (FR) is seen to have a positive relationship with maternal mortality. The coefficients of FR of 0.216308 and 0.356457 are positive and statistically significant at 0.05% level of significance for the MG and 0.01% level of significance for the PMG estimates respectively. Literature has long shown that in countries where fertility is high, maternal death rates and infant mortality rates are also high [62]. This is more pronounced in developing countries; women in these locations have significantly more pregnancies on average than women in industrialized nations, and their lifetime risk of pregnancy-related mortality is greater [63,64]. Increasing the use of family planning reduces pregnancy-related fatalities and population expansion through lowering fertility rates. Population Action International (PAI) published a report in 2015 which suggested that increased contraceptive prevalence in low-performing nations might cut maternal fatalities by 25% and relieve the strain on maternal health systems, allowing them to serve more women more effectively and efficiently [65,66]. 

GDP is found to be insignificant in both models. This also reinforces the scatter plot’s findings. This suggests that GDP does not influence maternal mortality in SADC countries. Some studies have found a positive association between GDP and maternal mortality. For instance, Ricthie (2016) [67] showed that maternal mortality fell as countries became wealthier. Another study found that the direct influence of GDP on health was positive but relatively modest [68]. When GDP is increases, the state receives more revenue through taxes and it will be able to improve its health expenditures. This will, then, improve health outcomes. However, Ensor, Cooper, and Davidson (2010) [69] point out that while it is self-evident that poorer nations have worse health results, the relationship between changes in wealth and health outcomes is less obvious at the national level. This might explain why the researchers were unable to find a link between GDP and maternal mortality in this study.

Results show that life expectancy (LE) has a negative relationship with maternal mortality. The coefficients of LE of −0.328755 and −0.372708 are negative and statistically significant at 0.01% level of significance for the MG and PMG estimators. These results support the earlier works of Hao et al. (2020) [70] and Okagbue (2020) [71] that postulated that maternal and paternal mortality ratios are both low in countries with high average life expectancy. Life expectancy is a useful measure of the status of health and living circumstances of populations, as well as the degree of development of health services. Having appropriate access to healthcare is linked to a decreased risk of death as people get older [70]. Maternal fatalities reduce where women have access to health care, increasing women’s life expectancy [72]. One of the leading causes of maternal mortality is a lack of access to health facilities and medical personnel. In Africa, there are currently 985 people per nurse/midwife and 3324 people for every medical doctor [73]. Many pregnant women do not obtain prenatal, delivery, or neonatal care, increasing their risk of death from severe bleeding, infections, or other problems.

Results show that HIV has a positive relationship with maternal mortality. The coefficients of 0.409207 and 0.543041 are positive and statistically significant at the 0.05% level of significance for the MG and at the 0.01% level of significance for the PMG estimators, respectively. These results support the earlier works [74,75,76,77] (Moran and Moodley, 2012; Alcorn, 2014: USAID, 2014 and Harvard Chan School, 2021). Pregnancy and HIV/AIDS both enhance a woman’s risk of contracting malaria, with potentially dangerous medication interactions obstructing successful treatment of both illnesses. HIV infection is linked to a variety of unfavourable health effects, including a higher chance of intrauterine infection [77]. 

A long-run association between maternal mortality and ICT is said to be present if the ECT is negative and significant. This is confirmed by the short-run results. In the short run, the results show that the speed of the adjustment parameter (0.432679 for the MG and −0.321992 for the PMG estimator) is negative and significant. For both models (MG and PMG), the ECT is negative and significant, and this confirms the existence of the long-run relationship between maternal mortality (MM) and the regressors that were used in this study.

### 4.5. Robustness Checks

The study performs an empirical model coefficients robustness test by using different estimation techniques (random effects (FE), fixed effects (FE), and GMM). Results are shown in Table 7 below.

The probability from the Hausman test is above 0.05 (0.215), indicating RE as a better choice compared to the RE model. On the dynamic models GMM (1) and GMM (2), the result of the Sargan test shows that the instruments employed in the model are valid given that the *p*-value of the Sargan test was 0.321.

ICT is found to be insignificant in all the static models (FE and RE). However, is it significant in all GMM models. The GMM is superior to the FE and RE models and its results confirm those of the PMG and MG models. It can thus be said that there is indeed a negative relationship between ICT and maternal mortality. This might support the argument that mobile technology provides unrivalled chances to improve maternal health outcomes. ICT, particularly mobile phones, may be an effective means of addressing health literacy issues and reaching women in underserved communities (Whittaker et al., 2012). FR is found to be significant in three models (RE, GMM (1) and GMM (2)). A positive relationship is found in all three models. This corroborates the PMG and MG’s results. GDP is found to be significant in the static models (FE and RE). The RE and FE found a negative relationship between GDP and maternal mortality. This implies that when GDP increases, maternal mortality drops. However, a superior estimation technique, GMM, showed that there is no relationship between GDP and maternal mortality. This corroborates the PMG and MG’s results. 

LE is found to be significant in the static models (FE and RE) and insignificant in the dynamic model (GMM). The results from the static model corroborate those from the PMG and MG estimators. The PMG and MG estimators showed that there is a negative relationship between LE and maternal mortality. This implies that countries with high life expectancy levels have lower maternal mortality rates. Life expectancy is a useful measure of the status of health and living circumstances of populations, as well as the degree of development of health services. According to research, having appropriate access to healthcare is linked to a decreased risk of death as people age [70]. Maternal fatalities reduce where women have access to health care, increasing women’s life expectancy [72].

Results also show that HIV has a positive relationship with maternal mortality in the FE and GMM (1) and GMM (2) estimators. The GMM (2) estimator used an alternative HIV prevalence rate. The GMM (2) used the HIV prevalence rate for both men and women. The study, then, reran its models with a GMM estimator (Model 2). The outcomes are largely similar with those presented in the FE and GMM (Model 1) analyses. The effects of HIV on maternal mortality remain robustly positive. This is consistent with literature. In countries with generalized HIV epidemics, for example, men are less likely than women to take an HIV test, less likely to access antiretroviral therapy, and more likely to die of AIDS-related illnesses than women [64].

## 5. Limitations 

The accessibility of quantitative data remains a serious concern. Data on some variables were not available. As mentioned in the literature section, this topic is affected by a range of geographical and sociocultural limitations. For example, literature shows that maternal mortality is higher in women living in rural areas and among poorer communities. However, obtaining data for the number of women living in rural areas is difficult. Furthermore, obtaining data on gender disparities on a macro level is a challenge. 

## 6. Conclusions and Recommendations

This study sought to examine the impact of ICT on maternal mortality in SADC countries. ICT has a crucial role in the alteration of healthcare services. The prospects for mobile phones to act synergistically with existing health systems in developing countries are numerous. Effective use of ICT in health can be a productive way to reduce health care costs and ameliorate the quality of care. Maternal mortality remains high in Africa, yet there are information communication technologies (ICTs) (such as the internet, mobile communication, social media, and community radios) that have the potential to make a difference. The literature identified some challenges that are faced by the health sector when using ICT. 

The use of ICT in reducing maternal mortality has been proven to be effective in this study. This shows that ICT has a high transformative potential, but SADC countries have not exploited this opportunity as shown by the marginal impact of ICT on maternal healthcare. It must be realized that the benefits from the adoption of ICT tools are not automatic. They depend on the ability and willingness of the government, the societies, and other relevant stakeholders to deploy and make complementary investments as well as offer technical support to ensure the contribution of ICT to health care. The challenges that women face when using ICT must be addressed. The fact that there are challenges means that millions of women are deprived of better sexual health information and medical care. The mobile gender gap should be closed (digital inclusion), mobile network connectivity boosted, and digital platforms must be created in order to empower women. Women who have lower access to health care, such as those living in rural regions, face a significant challenge in terms of awareness. Public awareness of digital health should also be addressed. Possible actions include improving digital health literacy at the population level; engagement of patients, families and communities; and education of patients about health.

## Figures and Tables

**Table 1 healthcare-10-00802-t001:** Compendium of eHealth projects in selected developing countries.

eHealth Project	Description	ICT Application	Country
**MOVE IT**	Pregnancies are registered, births are recorded, deaths are recorded, and the cause of death is recorded via a text message system.	Civil Registration and Vital Statistics	Ghana
**mCare**	Pregnancy registration and monitoring, as well as neonatal and post-partum care, have all benefited from the usage of mobile phone and database technology.	Data collection Community based healthcare	Bangladesh
**mUbuzima**	Community health workers (CHWs) utilize cell phones to give real-time data about community health indicators.	Health Information System	India
**RHEA**	Health information system to improve maternal and child care at the health centre level	Health Information System	Rwanda
**AMANECE**	Mobile phones are used to identify high-risk pregnancy warning signs and symptoms to assist primary health care providers in providing monitoring and follow-up for high-risk pregnancy cases and to enable prompt obstetric and newborn care treatments.	Patient monitoring Point-of-care support and decision support system	Mexico
**Pesinet**	Mobile-phone-based system monitoring information on mother and child health.	Patient monitoring Community based healthcare	Mali
**SMART**	Small battery-operated printers are used to receive and print early baby diagnosis test results in order to improve early infant diagnostic services by speeding up the delivery of results and determining treatment eligibility.	Diagnosis support	Cameroon Ethiopia Malawi Mozambique Zimbabwe
**Wawared**	Mobile technology solutions to enhance maternity and child care by increasing low-income pregnant women’s access to health services	SMS-based health education	Peru

Source: International Telecommunication Union (2012) [42].

**Table 2 healthcare-10-00802-t002:** Summary of Variable Description.

Variable ^1^	Description and Unit of Measurement	Source
MM	Maternal mortality ratio is the number of women who die from pregnancy-related causes while pregnant or within 42 days of pregnancy termination per 100,000 live births	World Bank
HIV	Prevalence of HIV, female is the percentage of females who are infected with HIV.	World Bank
ICT	Mobile cellular telephone subscriptions are subscriptions to a public mobile telephone service that provides access to the PSTN using cellular technology.	World Bank
FR	Fertility Rate. Total fertility rate represents the number of children that would be born to a woman if she were to live to the end of her childbearing years.	World Bank
GDP	Gross domestic product (GDP), total (2017 PPP $ billions).	World Bank
LE	Life Expectancy. Life expectancy at birth indicates the number of years a newborn infant would live if prevailing patterns of mortality at the time of its birth were to stay the same throughout its life.	World Bank

^1^ Fertility rate (FR), Economic Growth (GDP), Human Immune Virus (HIV), Information communication Technology (ICT), Life Expectancy (LE) and Maternal Mortality (MM).

**Table 3 healthcare-10-00802-t003:** Descriptive statistics.

	FR ^1^	GDP	HIV	ICT	LE	MM
**Mean**	4.3807	3.3078	7.1477	45.1157	58.2737	418.6071
**Median**	4.4130	9.9309	6.4500	33.3719	58.0365	433.0001
**Maximum**	6.7510	4.1456	24.2000	163.8752	77.8900	854
**Minimum**	1.3600	3.5032	0.1792	0.2486	44.5950	53
**Std. Dev.**	1.3964	7.3792	6.5320	42.5186	7.9601	196.4620
**Skewness**	−0.1775	3.5314	0.7440	0.9835	0.4392	−0.0756
**Kurtosis**	2.0825	15.0644	2.6297	3.1195	2.8888	2.1356
**Jarque–Bera**	10.8875	28.6624	26.4573	43.6965	8.8196	8.6620
**Probability**	0.0043	0.00000	0.00000	0.00000	0.01215	0.0131
**Sum**	1182.7912	8.9136	1929.90	12,181.26	15,733.90	113,024.0
**Sum Sq. Dev.**	524.578	1.4691	11,477.71	486,308.4	17,045.09	10,382,674
**Observations**	270	270	270	270	270	270

^1^ Fertility rate (FR), Economic Growth (GDP), Human Immune Virus (HIV), Information communication Technology (ICT), Life Expectancy (LE) and Maternal Mortality (MM).

**Table 4 healthcare-10-00802-t004:** Unit root tests.

Variable ^1^	Levin, Lin & Chu		Lm, Pesaran and Shin W-Stat	
	Stat.	Prob	Stat	Prob
MM	−0.4234	0.3112	−0.3384	0.3675
1st diff	−4.9406	0.0011	−5.6043	0.0000
LE	−0.6051	0.5257	0.14047	0.5559
1st diff	−3.6663	0.0001	−4.5141	0.0120
ICT	−3.8939	0.0000	−4.2341	0.0000
HIV	−0.7355	0.2310	−3.5033	0.0212
1st diff	−2.0847	0.0185	-	-
GDP	−0.5892	0.8501	1.2605	0.8963
1st diff	−3.6408	0.0162	−4.1661	0.0000
FR	−1.9429	0.0260	−2.3435	0.0096

^1^ Fertility rate (FR), economic growth (GDP), human immunodeficiency virus (HIV), information communication technology (ICT), life expectancy (LE) and maternal mortality (MM).

**Table 5 healthcare-10-00802-t005:** Pedroni cointegration test.

Cointergration Test	Intercept
**Test Statistics**	**Prob.**
Panel v-statistic	0.0009
Panel rho-statistic	0.0000
Panel PP-statistic	0.1083
Panel ADF-statistic	0.0210
Group Panel rho-statistic	0.0166
Group PP-statistic	0.5429
Group ADF-statistic	0.0120

**Table 6 healthcare-10-00802-t006:** MG and PMG results.

Variable	MG	PMG
Long-run coefficients		
ICT ^1^	−0.0750 ** (0.0358)	−0.0950 ** (0.0183)
FR	0.2163 ** (0.1044)	0.3564 *** (0.0760)
GDP	−0.4793 (0.3017)	−0.6936 (0.5696)
LE	−0.3287 *** (0.0483)	−0.3727 *** (0.1294)
HIV	0.4092 ** (0.0542)	0.5430 *** (0.0891)
Short-run coefficients		
ICT	−0.1451 * (0.1102)	−0.0342 ** (0.0174)
FR	0.1054 (0.0273)	0.2542 (0.0329)
GDP	0.0389 (0.0133)	−0.0342 *** (0.0174)
LE	−0.0195 * (0.0137)	−0.0005 ** (0.0485)
HIV	0.1286 ** (0.2492)	0.1734 ** (0.0936)
ECT ^2^	−0.4326 ** (0.1834)	−0.3219 ** (0.0915)

Note: *** 1% level, ** 5% level, and * 10% level. ^1^ Fertility rate (FR), economic growth (GDP), human immunodeficiency virus (HIV), information communication technology (ICT), life lxpectancy (LE) and maternal mortality (MM). ^2^ Error correction term.

**Table 7 healthcare-10-00802-t007:** Regression results. Dependent variable: MM.

Variable	FE	RE	GMM (1)	GMM (2)
**MM ^1^**			0.1333 * (0.0367)	0.0949 * (0.0238)
**ICT**	0.3245 (0.0179)	−0.0146 (0.0179)	−0.0765 *** (0.0118)	−0.0541 *** (0.0132)
**FR**	0.0013 (0.0088)	0.2447 ** (0.1738)	0.4765 *** (0.4275)	0.4133 *** (0.2069)
**GDP**	−0.1367 ** (0.2386)	−0.2447 ** (0.1738)	0.0003 (0.0000)	0.0020 (0.0007)
**LE**	−0.1522 *** (0.0247)	−0.1663 *** (0.0447)	−0.8057 (0.2261)	−0.3773 (0.0162)
**HIV**	0.0635 ** (0.4279)	0.1129 (0.0088)	0.3668 *** (0.0150)	
**HIVP ** ^2^**				0.4765 *** (0.42753)
**Observations** **Hausman Test** **Sargan test (*p*-value)**	270 0.2150	270 0.2150	270 0.321	270 0.321

^1^ Fertility rate (FR), economic growth (GDP), human immunodeficiency virus (HIV), information communication technology (ICT), life expectancy (LE) and maternal mortality (MM). ^2^ HIV prevalence rate—this is the percentage of people (both men and women) living with HIV. This was added in the second model (Model 2) for robustness purposes. *** 1% level, ** 5% level, and * 10% level.

## Data Availability

Not applicable.

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
