# Peer review of "ICT and Women’s Health: An Examination of the Impact of ICT on Maternal Health in SADC States"

_healthcare, 2022, doi:10.3390/healthcare10050802_

Round 1
Reviewer 1 Report
Dear author,
well done the paper is interesting, I had never red anything similar.
Potential instrument easy to realize able to reduce maternal mortality is crucial.
i have some minor comments:
try to limit abbreviations, to help the reader in understanding
The Abstract is the paper part that will make the difference for the reader to let him/her decide if to go trough the paper or not
there fore here some minor comment
1) I would have deleted the first sentence "
Attainment of sexual and reproductive health is regarded as a human rights matter. Not-11 withstanding this, " and go directly to..Maternal mortality
and would have done the same in the introduction
2) what do you mean by MG and PMG explain the abbreviations
3)I cannot understand if ICT has a positive or negative affect
due to the sentence
"
Results 23 showed that ICT has a negative effect on maternal health. This shows that ICT tolls contribute pos-24 positively to maternal health "
The paper in otherwise well written
I would have shortened the literature review
I would have mention the critical role that women awareness has for compliance with recommendations, these major interest is fetal well being, this come first, so if ICT technology may involve clear parts related to the connection of maternal well being to fetal well being pregnant women would be more keen in follow reccomendations.
I suggest to read and cite: PMCID: PMC8502344
Author Response
- I am afraid to say that we could not remove the first line (Attainment of sexual and reproductive health is regarded as a human rights matter. Notwithstanding this) because once we remove it, the meaning of the next 2 sentences would be reduced. We put that line because we want the reader to understand that sexual reproductive health is an important health issue and it is also a human right. Not offering better maternal health care is thus a violation of human right.
- The abbreviations for MG and PMG were written in full.
- ICT has a negative effect on maternal mortality. This means that when more people use ICT tools, maternal mortality decreases. On the other hand, it also means that ICT is contributing positively to maternal health (not maternal mortality). This means that the use of ICT is improving maternal health.
- We added some sentenced to incorporate the comment about critical role that women awareness plays.
Reviewer 2 Report
Thank you for giving me the opportunity to review the manuscript entitled: ICT and Women Health: An Examination of the Impact of ICT 2 on Maternal Health in SADC States, which aims to examine the impact of information and communication technologies on maternal mortality in SADC states through data available from the World Bank.
The article is well written, provides sufficient background information to back up and support the results. The references are correct. Here are some comments that may improve the quality of the article.
- In line 16 the abbreviation is repeated again (ICTs).
- Line 20-21, please check: “In light of 20 this this this study”
- On page 2 and 3, lines 97-100, instead of this study, Oluwaseun et al (2021) may be an alternative. Avoid repeating "this study".
- Page 3, line 114, check capital letter (Crosby, et. Al. 2017).
- Page 3, line 146, delete: “according to the study”.
- Page 3, line 147, “according to the studies”: which studies? There is no reference here.
- Page 4, line 170: check capital letters in text4baby.
- In table 1 of the introduction, you provide a table describing different projects to improve the health of the community, which is very interesting. Perhaps it would facilitate the reading of the introduction to include in this table the rest of the projects mentioned above: MobileVRS, Text4baby, Mobiles4Health and thus make it easier to read by summarising this section a bit.
- The inclusion of HIV in the model should be justified in the introduction.
- Page 8. add footnote to table 3 with the meanings of the initials.
- Please be consistent with the number of decimals. I don't think more than three are necessary
- The discussion of the results is not clear, it is done in the same results section. It would be useful for the discussion to be in a specific section.
- All tables have to be self-explanatory, add variable information, etc., at the bottom of the table.
- I do not know what ECT refers to in table 6.
- Point out the limitations of this study
- References should be corrected in a homogeneous formatting style, there are mistakes.
Author Response
- All the editorial issues were addressed.
- The inclusion of HIV in the model was justified under table 2.
- Page 8. add footnote to table 3 with the meanings of the initials. – the footnotes were added accordingly.
- The decimals were corrected for consistency. We managed to put 4 decimal places.
- Variable information were put on the main tables. Footnotes were added to explain the variables.
- A footnote to explain ECT was added. ECT is the error correction term.
Round 2
Reviewer 2 Report
I congratulate you on your work, I find it very interesting and it raises many challenges for the future. However, I must add that I believe that the issues I have raised in the first review have not been resolved, so prior to publication these should be considered.
I am missing a section on limitations in the discussion (limitations of your stuty, what problems did you find when doing this work). The discussion section as such is not clear either, although briefly it should be correctly identified. Please separate your results and the discussion you carry out in them and include it in a separate section, or mark that you carry out results and discussion together in the title. In addition, I keep finding decimals in the tables much higher than four digits (e.g. in table 3, with figures of 2, 4 and 5 decimals, in table 6 and 7) and also in the text they are not corrected. Add footnotes to tables in all tables that have abbreviations.
I still find the same errors in the format of the bibliographical references.
Author Response
Thank you so much for the valuable comments. We managed to incorporate all the comments you raised:
We added a section that outlined the limitations of the study. The section briefly explains the limitations that were encountered during the course of the research.
We also changed the title in section 4.5. the titled was changed to show that both the presentation and discussion of results were done at the same time.
We also made sure that all decimals have four digits.
We added footnotes on all tables with abbreviations.
Then for references, we will correct them when the paper has been accepted. There is a stage for proofreading. This is where all references will be corrected.